# Cost-analysis of COVID-19 sample collection, diagnosis, and contact tracing in low resource setting: The case of Addis Ababa, Ethiopia

**Amanuel Yigezu**[1]*, **Samuel Abera Zewdie**[2], **Alemnesh H. Mirkuzie**[1], **Adugna Abera**[3], **Alemayehu Hailu**[4], **Mesfin Agachew**[1], **Solomon Tessema Memirie**[5]

**1** National Data Management Center for Health, Ethiopian Public Health Institute (EPHI), Addis Ababa, Ethiopia, **2** Partnership and Cooperation Directorate, Ministry of Health, Addis Ababa, Ethiopia, **3** Parasitology Department, Ethiopian Public Health Institute (EPHI), Addis Ababa, Ethiopia, **4** Bergen Centre for Ethics and Priority Setting, Department of Global Public Health and Primary Care Medicine, University of Bergen, Bergen, Norway, **5** Addis Center for Ethics and Priority Setting, College of Health Sciences, Addis Ababa University, Addis Ababa, Ethiopia

* yigezuamanuel@yahoo.com

## Abstract

### Background

Ethiopia has been responding to the COVID-19 pandemic through a combination of interventions, including non-pharmaceutical interventions, quarantine, testing, isolation, contact tracing, and clinical management. Estimating the resources consumed for COVID-19 prevention and control could inform efficient decision-making for epidemic/pandemic-prone diseases in the future. This study aims to estimate the unit cost of COVID-19 sample collection, laboratory diagnosis, and contact tracing in Addis Ababa, Ethiopia.

### Methods

Primary and secondary data were collected to estimate the costs of COVID-19 sample collection, diagnosis, and contact tracing. A healthcare system perspective was used. We used a combination of micro-costing (bottom-up) and top-down approaches to estimate resources consumed and the unit costs of the interventions. We used available cost and outcome data between May and December 2020. The costs were classified into capital and recurrent inputs to estimate unit and total costs. We identified the cost drivers of the interventions. We reported the cost for the following outcome measures: (1) cost per sample collected, (2) cost per laboratory diagnosis, (3) cost per sample collected and laboratory diagnosis, (4) cost per contact traced, and (5) cost per COVID-19 positive test identified. We conducted one-way sensitivity analysis by varying the input parameters. All costs were reported in US dollars (USD).

### Results

The unit cost per sample collected was USD 1.33. The unit cost of tracing a contact of an index case was USD 0.66. The unit cost of COVID-19 diagnosis, excluding the cost for sample collection was USD 3.91. The unit cost of sample collection per COVID-19 positive

**Data Availability Statement:** All relevant data are within the paper and its Supporting Information files.

**Funding:** The author(s) received no specific funding for this work.

**Competing interests:** The authors have declared that no competing interests exist.

**Abbreviations:** ACEPS, Addis Center for Ethics and Priority Setting; COVID-19, Coronavirus disease-19; EPHI, Ethiopian Public Health Institute; FMoH, Federal Ministry of health; NDMC, National Data Management Center for health; PHEOCIMS, Public Health Emergency Operation Center Incidence Management Structure; USD, United States Dollar; WHO, World Health Organization.

individual was USD 11.63. The unit cost for COVID-19 positive test through contact tracing was USD 54.00. The unit cost COVID-19 DNA PCR diagnosis for identifying COVID-19 positive individuals, excluding the sample collection and transport cost, was USD 37.70. The cost per COVID-19 positive case identified was USD 49.33 including both sample collection and laboratory diagnosis costs. Among the cost drivers, personnel cost (salary and food cost) takes the highest share for all interventions, ranging from 51–76% of the total cost.

## Conclusion

The costs of sample collection, diagnosis, and contact tracing for COVID-19 were high given the low per capita health expenditure in Ethiopia and other low-income settings. Since the personnel cost accounts for the highest cost, decision-makers should focus on minimizing this cost when faced with pandemic-prone diseases by strengthening the health system and using digital platforms. The findings of this study can help decision-makers prioritize and allocate resources for effective public health emergency response.

## Background

On January 30, 2020, the World Health Organization (WHO) declared COVID-19 a public health emergency of international concern [1]. Globally, there were over 178 million confirmed cases of COVID-19, nearly 3.9 million deaths, and more than 163 million recovered cases reported as of 19th of June 2021. In Ethiopia, the total number of positive cases confirmed for COVID-19 reached nearly 275,000 with 4,276 deaths as of 19th of June 2021 [2].

Slowing down the transmission of the virus through a set of comprehensive strategies was crucial to reducing the morbidity and mortality from the virus. Central to these comprehensive strategies are core public health measures that break chains of person-to-person transmission, including (i) identification, isolation, testing, and clinical care for all cases, and (ii) tracing and quarantine of all contacts, and iii) use of face masks and implementation of social distancing measures [3, 4].

The public health measures emphasize the importance of early identification of incident cases to help reduce fatality [5]. Based on the epidemiology of the pandemic and capacity of testing in Ethiopia, individuals that fulfill the case definitions with acute respiratory illness (fever and at least one symptom of respiratory disease, i.e., cough, shortness of breath) and history of travel to or residence in a location reporting community transmission of COVID -19 during the 14 days prior to the onset of symptoms, were tested [6].

COVID-19 testing and contact tracing are thought to be the most effective in detecting and preventing transmission of the virus when the number of people to be tested or traced is small [7]. The volume of COVID-19 testing was inadequate in the early phase of the pandemic in Ethiopia, which could falsely reduce the number of people who would be isolated. Apart from reduced testing capacity, appropriate sample collection, turnaround time, cost, and sensitivity of the testing kits could play significant role in increasing the rate of COVID-19 detection [8]. To help improve COVID-19 case identification, in August 2020, Ethiopia implemented a community-based activity and testing program (COMBAT), a massive testing campaign against the novel coronavirus. The testing capacity has also improved following the purchase of more testing machines and locally produced COVID-19 test kits. By May 2020, more than half of the samples were collected from the community and health facilities, while specimen collection

from contact tracing was about 20%. The remaining samples were collected from quarantine and isolation centers, and airports for COVID-19 diagnosis [9].

Contact tracing during the COVID-19 pandemic helps in early detection and prompt isolation of new cases. Epidemiological models indicate that the efficacy of contact tracing and isolation is dependent on the community adherence and transmission dynamics. Contact tracing involves identification, listing, and follow-up of individuals who have contact with an infected person, two days before or 14 days after the onset of symptoms of a confirmed or probable case [10]. Ethiopia followed a more stringent COVID-19 contact tracing activities until October 2020 that was loosened afterwards when there was a wide spread community transmission where contact tracing is less effective [11]. Digital contact tracing was largely used in many countries. Digital contact tracing overcomes limitations that occur in traditional contact tracing related to delays in notification, identification of contacts in public gatherings, scalability, and recall errors [12, 13]. However, these technologies have little use in resource-limited areas due to poor infrastructure for information communication technology [14].

The global community, including Ethiopia, have invested a substantial amount of money in preventing and controlling the pandemic, including sample collection, diagnosis, and tracing of contacts of positive or suspected individuals. These investments are highly resource-intensive and cause more burden to low resource settings, like sub-Saharan Africa, as the health system is not resilient in these areas [15]. Different tools for estimating the financial resource requirements for the various interventions against COVID-19 were prepared [16–18]. However, the economic costs of COVID-19 testing and contact tracing have not been estimated. Knowledge about the costs for COVID-19 interventions is beneficial in mobilizing resources, planning, and budgeting by policymakers in these settings and for use in economic evaluations of different pandemic-related interventions. The information obtained from this study could inform resource prioritization for COVID-19 response and future epidemic/pandemic situations in low-income settings. In this paper, we estimated the unit cost and total cost for COVID-19 sample collection, diagnosis, and contact tracing for low-resource settings by evaluating the case of Addis Ababa, Ethiopia.

## Methods

### Study setting

The study was conducted in Addis Ababa, Ethiopia. Addis Ababa is the capital city of Ethiopia, with an estimated 5 million inhabitants [19]. In the city, there are 11 hospitals and 97 health centers. Addis Ababa has the highest concentrations of COVID-19 cases in the country [20]. The Ethiopian Public Health Institute (EPHI) is the technical wing for the Federal Ministry of Health (FMoH), which is responsible for leading the COVID-19 emergency response through its Public Health Emergency Operating Center (PHEOC). The PHEOC first started its COVID-19 response measures in Addis Ababa, as the city experienced high traffic within and outside the city. By May 2020, there were 46 COVID-19 testing laboratories nationally and four of them were located at EPHI. We used one of the four laboratories at EPHI to estimate the cost of COVID-19 testing, as all the laboratories follow the same procedure [21]. The COVID-19 samples were collected from the community, health facilities, quarantine centers, airports, and isolation centers, then transported to the COVID-19 testing laboratories. In addition, contact tracing activities were also centrally coordinated by PHEOC.

### Costing approach

We conducted the costing from the healthcare system perspective. We used a combination of micro-costing (bottom-up) and top-down approaches to estimate resources consumed and the

unit costs of the interventions. Activities under each service were defined, measured, and valued. We estimated the costs by listing action items for each intervention, describing the specific resources needed to implement the intervention, and assigning costs to all the resources to account for the opportunity costs of the interventions. We classified the costs into capital and recurrent costs. Capital costs include buildings, equipment, and vehicles. Recurrent cost includes supplies and personnel.

Both primary and secondary data were used to estimate the costs of COVID-19 sample collection, COVID-19 diagnosis, and contact tracing. The data were collected from EPHI through the review of financial records, and expert consultations. Consultations with EPHI experts were conducted to estimate the supplies required to collect COVID-19 samples and conduct contact tracing. Supplies used for the COVID-19 laboratory diagnosis were collected from the COVID-19 laboratory unit of the EPHI, which has been conducting COVID-19 diagnosis since the pandemic began. The institute provides a meal for the laboratory personnel. Resource use for personnel, including salary, allowances, and meals, were collected from the head of the laboratory and the finance directorate within the EPHI. Equipment used for the COVID-19 laboratory was collected from the laboratory, and the price of the equipment and supplies were collected from manufacturers' website and the Ethiopian Pharmaceutical and Supply Agency (EPSA). Equipment was annualized using a discount rate of 3% with an assumed lifespan of 5 years [22].

The institute provides transportation services for contact tracing activities, sample collection and to laboratory personnel conducting COVID-19 diagnosis. The vehicles were either owned by the institute, supplied for COVID-19 activities from different agencies, or rented from private organizations. We used rental values for all the vehicles within the institute, collected from the finance directorate within EPHI. The number of vehicles allocated to COVID-19 sample collection, diagnosis and contact tracing were collected from the transportation unit at EPHI. We used averaged rental value to estimate building costs. The rental value for building (per meter square per month) was taken from local experts who engage in activities related to rental services.

To estimate the cost per outcome; service outcome measures were taken from the laboratory conducting the COVID-19 diagnosis, from District Health Information System (DHIS), and daily COVID-19 updates by the EPHI. We collected data on the following outcomes: the number of COVID-19 samples collected, the number of laboratory diagnoses performed, the number of COVID-19 contacts traced, and the number of positive COVID-19 tests identified. We report the cost for the following outcome measures: (1) cost per sample collected, (2) cost per laboratory diagnosis, (3) cost per sample collected and laboratory diagnosis, (4) cost per contact traced, and (5) cost per COVID-19 positive test identified. We only considered cost of sample collection and testing to estimate the cost per COVID-19 positive test and it does not include the cost of contact tracing since some of the samples were collected directly (i.e. individuals with COVID-19 symptoms opting for testing) and not through contact tracing.

We conducted a one-way sensitivity analysis on the cost of identifying COVID-19 cases through laboratory diagnosis by varying the values of low and high input parameters. The lower and the higher value choice were made considering the clinical and economic feasibility of the range concerning the setting and consensus among experts [23]. All costs were collected in Ethiopian Birr (ETB) and then converted to USD (1 USD = 35.55 ETB based on the average exchange rate from the 1st of March to the end of December 2020) [24].

### Ethical consideration

Ethical clearance was acquired from the Ethiopian Public Health Institute scientific and ethical review committee (EPHI-IRB-275-2020). The data is fully anonymized and informed consent from patients was not required.

## Results

### Unit cost of sample collection, contact tracing, and laboratory diagnosis

The total number of COVID-19 samples collected in Addis Ababa from June to December 2020 were 598,502 with 68,578 confirmed cases with a total cost of USD 797,397. Over the same period, 1,070,686 COVID-19 contacts were traced, costing USD 706,992. From May to December 2020, the COVID-19 laboratory conducted 73,955 COVID-19 DNA-PCR tests with 7,668 confirmed positive samples at the cost of USD 289,077. The monthly trends of the unit costs of sample collection, contact tracing, and testing, as shown in Fig 1, indicate that the unit cost of laboratory diagnosis was the highest compared to sample collection and contact tracing. The unit costs were higher in the first month of the interventions. The average unit costs of the sample collection, contact tracing, and laboratory diagnosis were USD 1.33, 0.66, and 3.91, respectively.

For sample collection, the cost of personnel takes the highest share of the total cost (around 56%), followed by vehicle cost (38%), while supplies cost contributes to only 6% of the total cost. However, the cost of building is insignificant when compared to other sample collection cost. From the total vehicle cost of USD 305,632.95, about 70% and 30% were spent to transport samples from the community and from health facilities to COVID-19 testing laboratories, respectively.

Once the laboratory received the samples, it conducted a reverse-transcription polymerase chain reaction diagnosis for COVID-19. As presented in Fig 2, for COVID-19 laboratory diagnosis, the cost of personnel was about 76% of all costs. The cost of supplies (15%) took the second-highest share of the total cost.

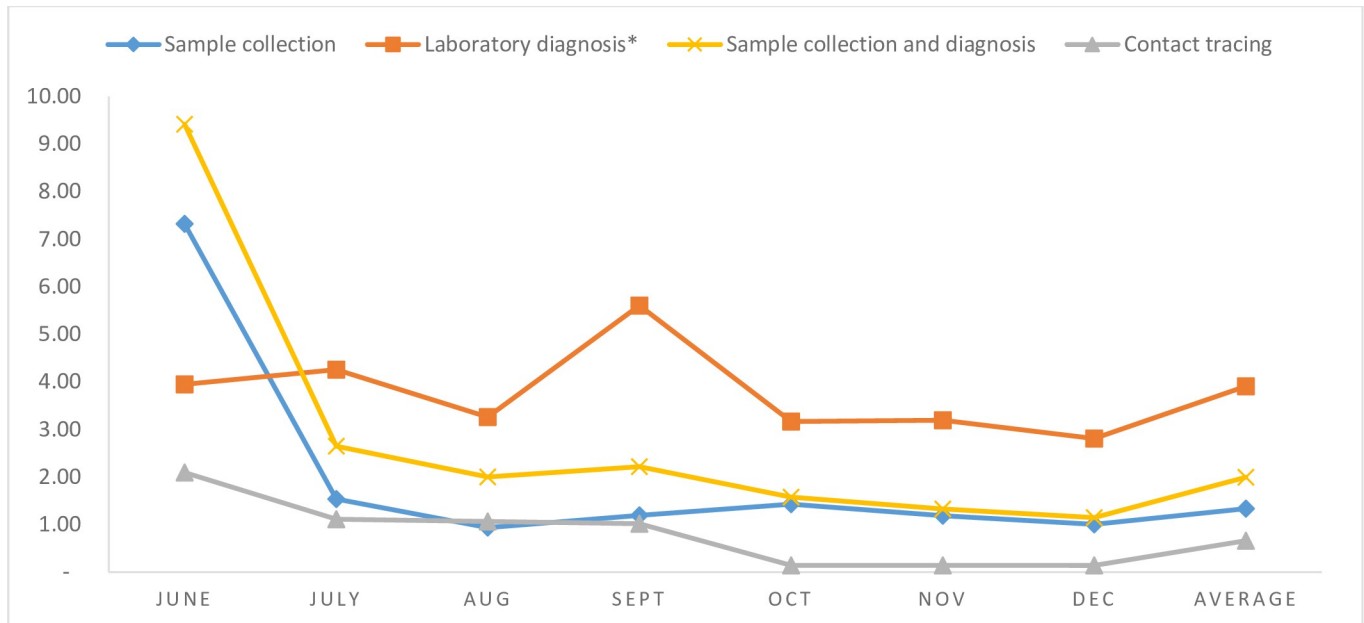

**Fig 1. The unit costs of COVID-19 sample collection, contact tracing, and diagnosis over the study period.** * The cost of laboratory diagnosis does not include the costs of sample collection.

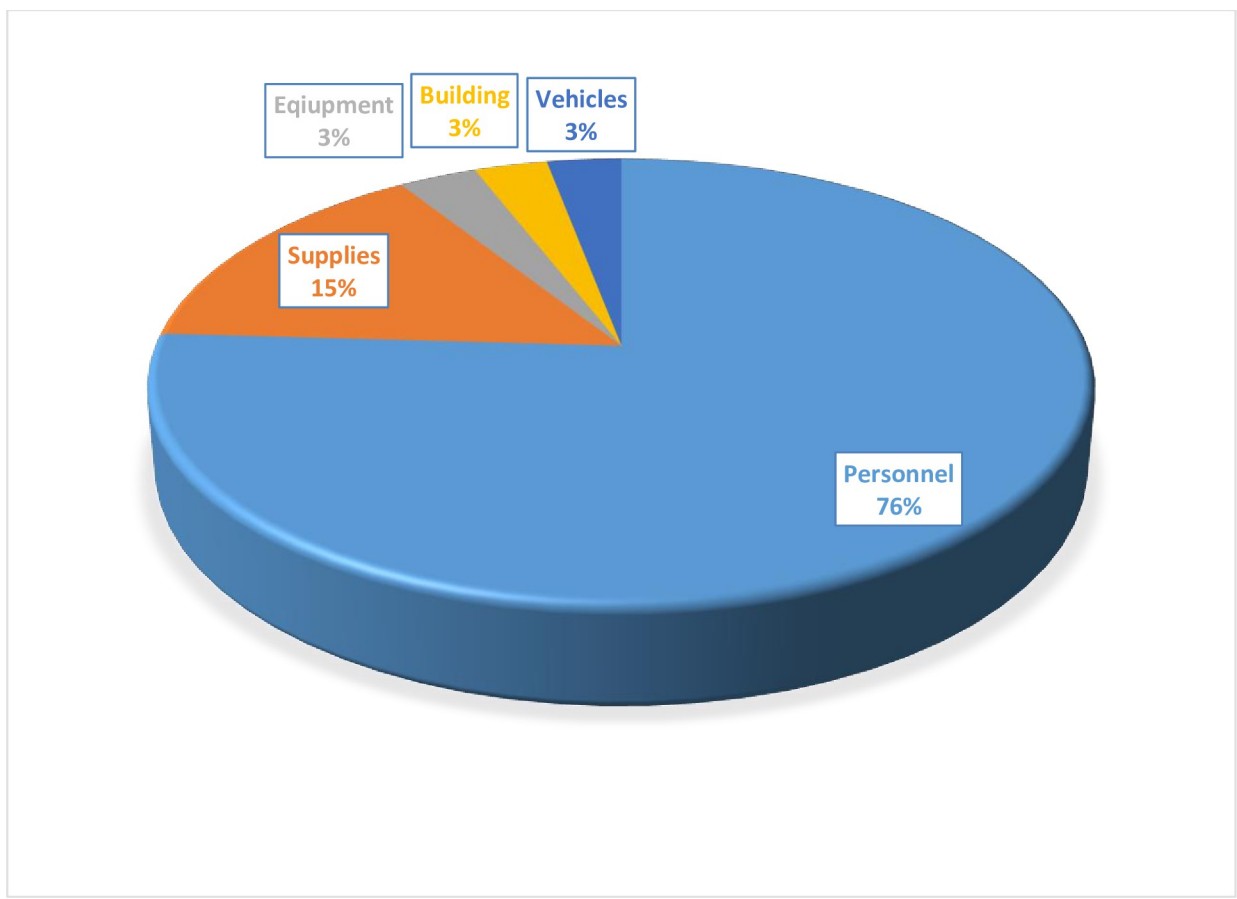

**Fig 2. Percentage share of costs for COVID-19 laboratory diagnosis, Addis Ababa, 2020.** *Personnel: food cost (16%), Salary (60%).

The total cost of COVID-19 laboratory diagnosis, including sample collection, is USD 5.26 per COVID-19 test. The sample collection accounts for about 25% of the total cost. The cost of tracing a COVID-19 contact individual is USD 0.66. About 44% of this cost was attributed to transportation and 51% to personnel costs. The cost of supplies, building, and equipment constitute less than two percent of the total cost of COVID-19 contact tracing.

## The cost of identifying COVID-19 positive cases

The cost of COVID-19 sample collection to identify one COVID-19 positive individual is USD 11.63. After receiving the samples, the cost per COVID-19 positive identification was USD 37.70. The cost to identify COVID-19 positive individuals including the cost of sample collection is USD 49.33. Out of the total cost of sample collection and diagnosis, the cost of sample collection takes about 24% of the total cost, while laboratory diagnosis after receiving the sample accounts for about 76% of the total cost. The cost of identifying COVID-19 positive cases through contact tracing is USD 54.00 which ranges from USD 520 in June 2020 to USD 18 in December 2020 (Fig 3).

## Sensitivity analysis results

We conducted a one-way sensitivity analysis on the discount rate and useful life years used to estimate capital costs, vehicle and building rental costs, equipment and supply costs, and

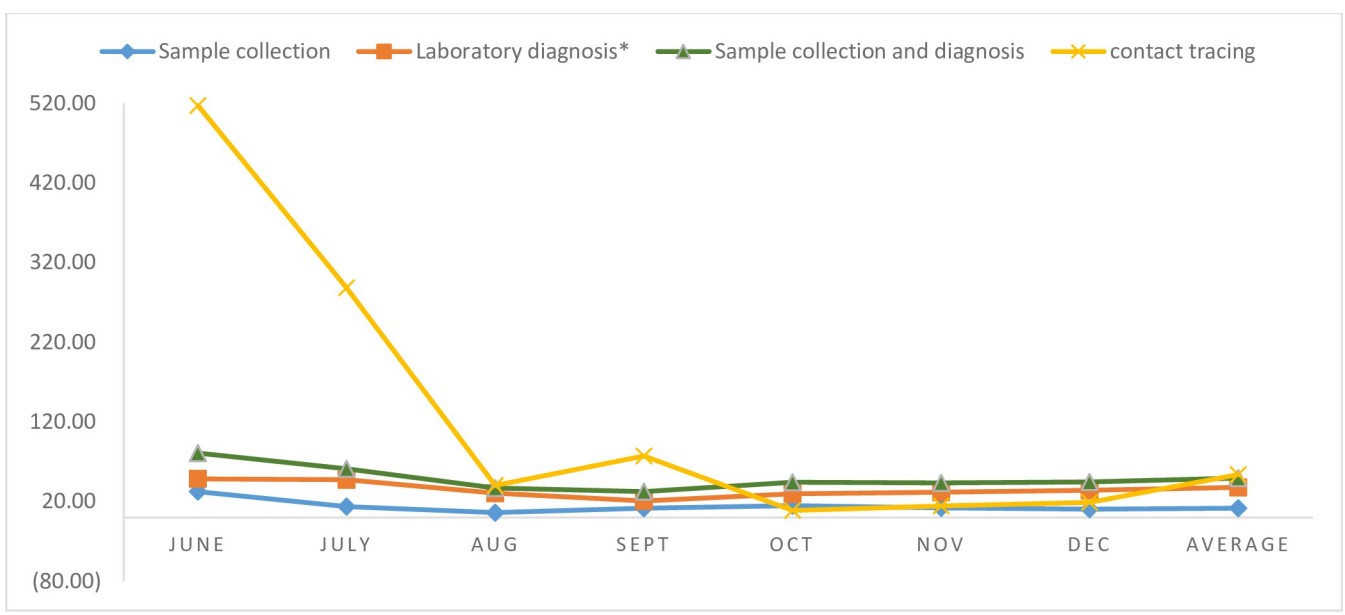

**Fig 3. Unit cost of sample collection and laboratory diagnosis to identify COVID-19 positive cases.** *Laboratory diagnosis cost only includes the cost after receiving the COVID-19 samples.

personnel costs, including food, salary, and allowance payments. We used a 20% variation of the parameters to assess its impact on the cost of identifying COVID-19 positive cases after the samples are received in the laboratory. The one-way sensitivity analysis showed that the COVID-19 positivity rate is the most important parameter followed by risk allowance payment for COVID-19 workers. Next to the risk allowance, food costs, salary of the laboratory experts, and supply costs are sensitive. Fig 4 below presents the impact of different changes in the input parameters on the costs of COVID-19 laboratory diagnosis.

## Discussion

This study estimated the costs of COVID-19 sample collection, diagnosis, and contact tracing. The unit costs were USD 1.33, 0.66, and 3.91 for sample collection, diagnosis, and contact tracing, respectively. Our study found that personnel cost was the main driver of the cost for sample collection, diagnosis, and contact tracing. Vehicle cost was also the cost driver for sample collection and contact tracing. The sensitivity analysis on the input parameters of the COVID-19 diagnosis indicates that the COVID-19 positivity rate and risk allowance payment to health professionals highly impact the unit cost of identifying a COVID-19 positive individual.

Our unit cost estimate for sample collection and diagnosis of COVID-19 was USD 5.24. Of that, USD 1.33 was spent on sample collection, and USD 3.91 was accrued to diagnose COVID-19 after receiving the samples to a COVID-19 laboratory unit. These costs might vary across countries according to the type of testing and resources used [25, 26]. The cost to identify a COVID-19 positive individual was USD 49.33, of which sample collection contributed to USD 11.63, and the cost to identify a COVID-19 positive individual after receiving the sample to the laboratory is about USD 37.70. As Ethiopia scaled up its testing capacity, efficient resource allocation to COVID-19 sample collection and testing would benefit the health system. Scaling up testing capacity is cost-efficient as it averts hospitalizations due to COVID-19 disease [26].

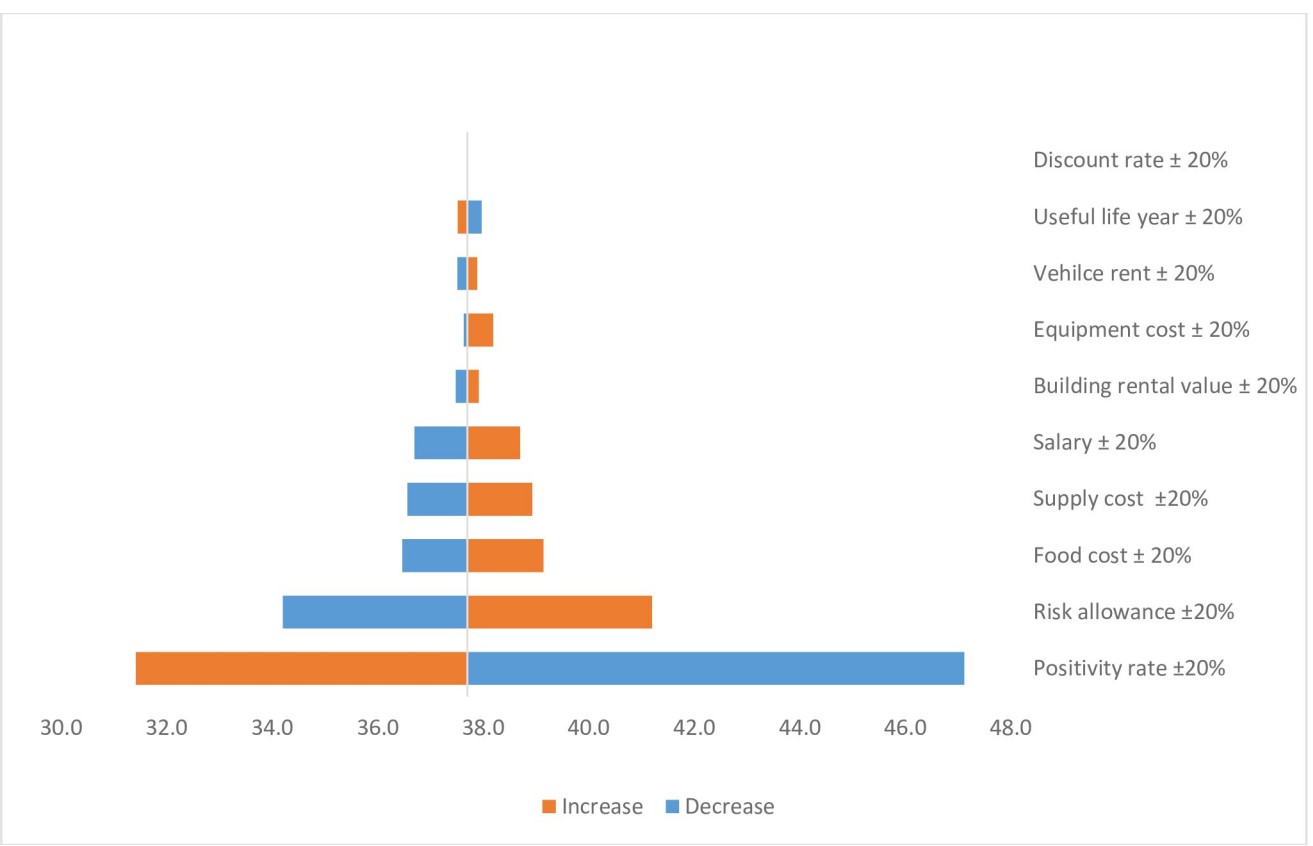

**Fig 4. One-way sensitivity analysis on the cost of identifying COVID-19 positives through laboratory diagnosis.** N.B: the cost includes only resource use after the sample is transported to the laboratory unit.

For COVID-19 sample collection, 56% of the costs were spent on personnel, and 38% was spent on vehicles. As the number of COVID-19 testing laboratories is limited, the costs associated with sample collection bore a high burden to the health sector because of the distance travelled to transport the samples and personnel related costs. These costs contribute to a third of the total laboratory testing cost, indicating the health system might avert a large amount of investment had there been access to nearby laboratory facilities that conduct COVID-19 testing.

Share of the costs at the COVID-19 laboratory after receiving samples were about 76% for personnel (60% salary/allowance and 16% food) and 15% for the cost of supplies. Allowance payment and meal price that were not usual cost components in the health system now take a significant amount of the resources. Allowance payment for the health workers on COVID-19 takes more than fivefold of their average salary. The sensitivity analysis also indicates that the unit cost of identifying COVID-19 cases is determined mainly by the COVID-19 positivity rate and payment of risk allowance for personnel. If the health system is stronger and able to include COVID-19 laboratory testing in the routine health system, we could save resources that are invested as risk payment for health professionals. This might require a resilient health system that can effectively and efficiently tackle epidemics/pandemics and, therefore, the need for health system strengthening.

The cost of tracing a contact of COVID-19 infected individual was USD 0.66; 51% of which was personnel salary and allowance, followed by costs of vehicles, which was about 44% of the

total cost. As we considered the vehicle costs in rental value, the costs associated with transportation might be over-estimated. Our cost estimate is less than the unit cost estimate from a previous study in South Africa, which estimated the cost per contact to be USD 2.67 in their context. This variance in cost may be due to differences in the methods used to estimate the unit costs [27]. In Ethiopia, however, it costed about USD 54.00 per COVID-19 positive case identified through contact tracing. The cost per COVID-19 positive case identified was very high in the first few months when the pandemic was not widely distributed in the community. Improved contact tracing methods, like the use of digital contact tracing, would also reduce the cost of tracing a COVID-19 suspect. This would also increase the efficiency and quality of the contact tracing activities by improving the identification of contacts in public gatherings and reducing recall errors [13].

As the community distribution of the pandemic increases, the total cost of testing and contact tracing will be very high as there would be more COVID-19 suspects who may have contracted the virus. However, the efficacy of these two interventions would decrease [7]. Therefore, the economic benefit of testing and contact tracing of COVID-19 suspects would be less compared to their implementation in the early phase of the pandemic. As this study is only aimed at estimating the costs of tracing a COVID-19 contact and testing, we recommend further studies that combine the costs relative to the effects for different sets of interventions during different phases of the pandemic.

The introduction of vaccines can vastly reduce the transmission and thereby the economic costs associated with other COVID-19 interventions, including sample collection, testing, and contact tracing [28, 29]. Ethiopia has recently started vaccinating its community against COVID-19. It is estimated that about USD 382.3 million is needed for the COVID-19 vaccine procurement, delivery, and health system strengthening [30]. As of May 2021, about 2 million people were tested for COVID-19 in Ethiopia. The findings from this study will provide input to estimating the potential cost savings that arise from the introduction of vaccines in low-resource settings, including Ethiopia. For example, the direct healthcare impact of resource diversion from other essential health services would have a tremendous economic impact on the health system [31, 32]. Therefore, we recommend further studies to assess the opportunity cost of interventions against COVID-19 on the health system in low and middle-income countries.

The response against COVID-19 pandemic has disrupted the health system of countries. Despite making remarkable progress in achieving the millennium development goals, the COVID-19 may hamper developing counties' progress towards the Sustainable Development Goals (SDG) [33]. Ethiopia's per capita spending was $33.2 in 2016/17. It is estimated that additional per capita spending of USD 41 is needed each year to make progress toward SDG in low and middle-income countries [34]. The resource diversion from the already weak per capita health spending to COVID-19 response might even cancel the MDG era's achievements. To protect the health of the society, maintaining essential health services was a significant recommendation especially in a resource-limited setting that could be overburdened by the COVID-19 pandemic [35]. To recommence and keep the non-COVID-19 essential services in pace, Ethiopia has put approaches for delivering those services [36].

This study is not without limitations. First and foremost, the available data for contact tracing and sample collection from March-May 2020 were not complete. Therefore, we included data for laboratory diagnosis starting from the month of May while data for contact tracing and sample collection from June-December 2020. This may result in incomplete cost information. In addition, since the application of COVID-19 activities was changing in its approach, we might have missed capturing all of the cost at each level due to recall bias and poor recording of the expenses associated with the COVID-19. For example, most of the contact tracing

activities before September were intense. Because of multiple administrative units within the EPHI, it was difficult to estimate and allocate utilities such as electricity, water, and telecommunication costs. Moreover, our study did not consider the possibility that samples might be retested again. This might also underestimate the unit cost estimate of our findings. This study did not consider the costs of other major interventions, including the costs of quarantine centers, establishing and expansion of COVID-19 treatment centers and diagnostic facilities, enforcement of COVID-19 prevention and infection prevention and control (IPC) measures, and promotion of disease prevention and control. Furthermore, as this study focus in Addis Ababa, we recommend additional studies to be conducted at the national level.

## Conclusion

This study attempted to estimate the cost of COVID-19 sample collection, testing, and contact tracing in a low-income country, such as Ethiopia. The findings indicate that the cost of interventions against pandemic-prone disease could be a substantial burden to the health system in such settings. Strategies that support efficient use of resource when faced with a pandemic-prone disease are crucial. This study will support the allocation of resources within the health system in low-income countries by identifying feasible and justifiable areas to reduce costs. We found that the personnel cost for COVID-19 testing and contact tracing is high due to additional payment for risk allowance and food costs and salary. The unit cost estimates for COVID-19 sample collection, testing and contact tracing could be used as an input by researchers for conducting economic evaluation of COVID-19 interventions. In addition, it may help policy and decision makers for planning and budgeting on future pandemic response and control measures.

## Supporting information

**S1 Appendix.**
(DOCX)

## Acknowledgments

We would like to thank the staff of the Ethiopian Public Health Institute (EPHI) for their support on this study.

## Author Contributions

**Conceptualization:** Amanuel Yigezu, Samuel Abera Zewdie, Alemnesh H. Mirkuzie, Solomon Tessema Memirie.

**Data curation:** Amanuel Yigezu, Samuel Abera Zewdie, Solomon Tessema Memirie.

**Formal analysis:** Amanuel Yigezu.

**Investigation:** Amanuel Yigezu.

**Methodology:** Amanuel Yigezu.

**Project administration:** Amanuel Yigezu.

**Resources:** Amanuel Yigezu.

**Supervision:** Amanuel Yigezu, Solomon Tessema Memirie.

**Validation:** Amanuel Yigezu.

**Visualization:** Amanuel Yigezu.

**Writing – original draft:** Amanuel Yigezu.

**Writing – review & editing:** Amanuel Yigezu, Samuel Abera Zewdie, Alemnesh H. Mirkuzie, Adugna Abera, Alemayehu Hailu, Mesfin Agachew, Solomon Tessema Memirie.

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
