## [Decision Letter · Decision Letter 0]

11 Oct 2021

PONE-D-21-20233

Cost-analysis of COVID-19 sample collection, diagnosis, and contact tracing in low resource setting: the case of Addis Ababa, Ethiopia

PLOS ONE

Dear Dr. Amanuel Yigezu,

Thank you for submitting your manuscript to PLOS ONE. After careful consideration, we feel that it has merit but does not fully meet PLOS ONE’s publication criteria as it currently stands. Therefore, we invite you to submit a revised version of the manuscript that addresses the points raised during the review process.

We look forward to receiving your revised manuscript.

Kind regards,

Carlos Alberto Zúniga-González, Ph.D

Academic Editor

PLOS ONE

Additional Editor Comments (if provided):

Dear author, the main concern with the paper is that it is very vague on the sources of the data, making it hard both to replicate and to get a sense of how much the results might be able to generalize, say, across Ethiopia. I therefore recommend that the authors provide more specific information about data sources, perhaps in a table form, to enable replication to the greatest extent possible.

I suggest this references:

1) Dios-Palomares, R ; Alcaide, D ; Diz, J ; Jurado, M ; Prieto, A ; Morantes, M ; Zuniga, C.A. (2015). Analysis of the Efficiency of Farming Systems in Latin America and the Caribbean Considering Environmental Issues REVISTA CIENTIFICA-FACULTAD DE CIENCIAS VETERINARIAS, Volume: 25 Issue: 1 Pages: 43-50 https://publons.com/publon/3106827/

2) Zuniga González, C. A. (2020). Total factor productivity growth in agriculture: Malmquist index analysis of 14 countries, 1979-2008. Revista Electrónica De Investigación En Ciencias Económicas, 8(16), 68–97. https://doi.org/10.5377/reice.v8i16.10661

3) Blanco-Orozco, N. V., Arce-Díaz, E., & Zúñiga-Gonzáles, C. (2015). Integral assessment (financial, economic, social, environmental and productivity) of using bagasse and fossil fuels in power generation in Nicaragua. Tecnología En Marcha Journal, 28(4), pág. 94–107. https://doi.org/10.18845/tm.v28i4.2447

Reviewers' comments:

Reviewer's Responses to Questions

**Comments to the Author**

1. Is the manuscript technically sound, and do the data support the conclusions?

Reviewer #1: Partly

Reviewer #2: Partly

2. Has the statistical analysis been performed appropriately and rigorously? 

Reviewer #1: I Don't Know

Reviewer #2: Yes

3. Have the authors made all data underlying the findings in their manuscript fully available?

Reviewer #1: No

Reviewer #2: No

4. Is the manuscript presented in an intelligible fashion and written in standard English?

Reviewer #1: Yes

Reviewer #2: Yes

5. Review Comments to the Author

Reviewer #1: Major comments and questions:

Overall: Thank you for the opportunity to review this manuscript. I appreciate the authors’ contributions with this important study to the field of public health and public health economics specifically. The study provides a straightforward analysis of the costs associated with covid-19 sample collection, diagnosis, and contact tracing; however, I have a few considerations to improve the manuscript prior to it being accepted for publication.

1. The introduction section provides a detailed background and justification for the study, and is well written.

2. The methods section needs to be improved as there is not enough detail of the costing approach. First, please clarify the actual study perspective. You list that it is a healthcare provider perspective, but you include more than just provider-related costs. Provider costs are only relevant to the staff providing services and typically only includes salary and time. I believe your study is an actual healthcare sector or system perspective as you include direct provider and payer costs related to sample collection, diagnosis, and contact tracing, not just those costs directly related to the provider. The health sector perspective includes all providers and payers and assumes that all medical costs are paid by the health sector. Second, I do not know what costing method you used in the study. You stated that “activities under each service were defined, measured, and valued. We estimated the costs by listing action items for each intervention, describing the specific resources needed to implement the intervention, and assigning costs to all the resources based on opportunity costs used for the intervention”. Did you rely on a macro-costing method, activity-based costing, or micro-costing for this? Please clarify this and provide more information on your collection procedures. Also, it would be helpful to include your cost inventory as an appendix or supplementary file to review all of the resources under each activity that are included in your costs. You’ve described in general what was included, but I still don’t know what specifically makes up all of the resources and subsequent costs in the study. Third, I am unsure what makes up the cost for identifying a positive Covid-19 case? I would assume that the costs of sample collection, laboratory diagnosis, and contact tracing are included, but aggregating the costs of these three from your results does not add up to the costs for identifying a positive case in the text. The details for this are missing in the methods and need to be included.

3. The results section has some inconsistencies. Figure 1: In the text you state that this data is from May – December; however the figure only shows June – December. Please add footnotes for figure 1 to explain why data for May is not shown on the figure. Also, for figure 1, you have added “sample collection and diagnosis” as a result to share. You only state throughout the text that you are assessing the three outcomes independently (1. sample collection, 2. Laboratory diagnosis, and 3. Contact tracing). What is the value added to show us sample collection and diagnosis combined?

Figure 2: Percentages are better visualized in a pie chart or donut chart, or a stacked bar chart, since you are trying to show proportions out of a whole. The way it is presented here would draw immediate attention to the x-axis scale where we see it does not total 100%. For figure 2 I also recommend adding footnotes that provide more details of the items/resources that are included in the costs for those categories. I have seen other studies provide this as the general perspective is the table should provide enough detail to stand alone.

4. The discussion is written well and informative. My only suggestion it to add more about how Ethiopia can use these results for budgeting and priority planning and speak to the benefit of the study for future economic evaluation.

Additional Comments:

1. The references with website links need to be reviewed. I received an error for the links for references #1, 17, and 18 when attempting to open.

Reviewer #2: My main concern with the paper is the non-transparency of the data. The costs are clearly communicated, but it is oftentimes unclear where the data comes from. Further, partly because it is unclear where the data come from, it is also unclear whether the results are expected to be representative for Ethiopia as a whole, or just Addis Ababa. And are different components of the total cost estimates more/less representative of Addis Ababa or Ethiopia?

The study should be replicable to the greatest extent possible, and right now, the sources of the data are so vague that it would be very hard for someone else to replicate the study. For instance, it says in the first paragraph under the section "Costing approach" that "The data was collected from the EPHI through the review of financial records, literature, and expert consultation." This is very vague -- what data came from what source, and what were the sources more precisely? And, another example: further down in the same section, rental values are discussed, but where does the rental value data come from? It says it was collected from "local experts who engage in activities related to rental services." What does that mean, and how can I find that data?

6. PLOS authors have the option to publish the peer review history of their article (what does this mean?). If published, this will include your full peer review and any attached files.

Reviewer #1: No

Reviewer #2: No

---

## [Author Response · Author response to Decision Letter 0]

10 Feb 2022

January 29, 2022

Dr. Sanjay Kumar Singh Patel

Academic Editor

PLOS ONE

Dear Editors,

We are submitting our revised manuscript entitled "Cost-analysis of COVID-19 sample collection, diagnosis, and contact tracing in low resource setting: the case of Addis Ababa, Ethiopia " (PONE-D-21-20233) to PLOS ONE. 

We would like to thank the editor and reviewers for the constructive comments and suggestions which have helped to strengthen our work substantially. 

We have now included “Inputs and data sources” section in the supporting document to provide specific information about the data sources that we have used and to enable replication of our study findings to the greatest extent possible. 

Below, we have provided a detailed point-by-point response to all comments and suggestions raised in the review.

Thank you very much for your consideration of our manuscript.

Sincerely,

Amanuel Yigezu, on behalf of the authors

 

Point-by-point response to the reviewers’ comments

Reviewer #1:

The introduction section provides a detailed background and justification for the study, and is well written.

Authors’ response: We thank the reviewer for his/her positive comments on our manuscript and the additional comments are addressed below.

The methods section needs to be improved as there is not enough detail of the costing approach. First, please clarify the actual study perspective. You list that it is a healthcare provider perspective, but you include more than just provider-related costs. Provider costs are only relevant to the staff providing services and typically only includes salary and time. I believe your study is an actual healthcare sector or system perspective as you include direct provider and payer costs related to sample collection, diagnosis, and contact tracing, not just those costs directly related to the provider. The health sector perspective includes all providers and payers and assumes that all medical costs are paid by the health sector.

Authors’ response: Thank you for your comments. We agree with the reviewer that it is a health care system perspective and currently, it reads as “health care system perspective” on the relevant sections of the paper (line 33 and 127). Further more, we have now improved the methods section to clarify on the costing approach. Please see under Methods, “Costing approach” (line 127-129).

Second, I do not know what costing method you used in the study. You stated that “activities under each service were defined, measured, and valued. We estimated the costs by listing action items for each intervention, describing the specific resources needed to implement the intervention, and assigning costs to all the resources based on opportunity costs used for the intervention”. Did you rely on a macro-costing method, activity-based costing, or micro-costing for this? Please clarify this and provide more information on your collection procedures. Also, it would be helpful to include your cost inventory as an appendix or supplementary file to review all of the resources under each activity that are included in your costs. You’ve described in general what was included, but I still don’t know what specifically makes up all of the resources and subsequent costs in the study.

Authors’ response: Thank you for raising this important point. We have now included the following sentence under Method, Costing approach, lines 127-129: “We used a combination of micro-costing (bottom-up) and top-down approaches to estimate resource consumed and the unit costs of the interventions”. We have also included a cost inventory in the supporting document. We have also modified the section describing our data collection procedure (Line 137-157).

Third, I am unsure what makes up the cost for identifying a positive Covid-19 case? I would assume that the costs of sample collection, laboratory diagnosis, and contact tracing are included, but aggregating the costs of these three from your results does not add up to the costs for identifying a positive case in the text. The details for this are missing in the methods and need to be included.

Authors’ response: Thank you for raising this point. We chose not to include the cost of contact tracing since not all samples are collected through contact tracing. We therefore used only (sample collection and lab diagnosis to inform the cost per COVID-19 test and the cost per COVID-19 positive identification. We have now included this clarification in the method section “We only considered cost of sample collection and testing to estimate the cost per COVID-19 positive test; it does not include the cost of contact tracing since some of the samples are collected directly (i.e. from individuals with COVID-19 symptoms opting for testing) and not through contact tracing.”

The results section has some inconsistencies. Figure 1: In the text you state that this data is from May – December; however the figure only shows June – December. Please add footnotes for figure 1 to explain why data for May is not shown on the figure.

Authors’ response: Thank you for your comment. Even though data were collected for the period May to December, complete data for May were not available for contact tracing and sample collection. However, data were available for laboratory diagnosis from May to December. We have now added the following under the limitation section: Line 294-296 “This study is not without limitations. First and foremost, the available data for contact tracing and sample collection from March- May 2020 were not complete. Therefore, we included data for laboratory diagnosis starting from the month of May while data for contact tracing and sample collection from June-December, 2020”. 

Also, for figure 1, you have added “sample collection and diagnosis” as a result to share. You only state throughout the text that you are assessing the three outcomes independently (1. sample collection, 2. Laboratory diagnosis, and 3. Contact tracing). What is the value added to show us sample collection and diagnosis combined?

Authors’ response: Thank you for the comment. We wanted to add this information to inform the combined cost of COVID-19 sample collection and laboratory diagnosis. To clarify the point we have added the following description: “We reported the cost for the following outcome measures: (1) cost per sample collected, (2) cost per laboratory diagnosis, (3) cost per sample collected and laboratory diagnosis, (4) cost per contact traced, and (5) cost per COVID-19 positive test identified.”

Figure 2: Percentages are better visualized in a pie chart or donut chart, or a stacked bar chart, since you are trying to show proportions out of a whole. The way it is presented here would draw immediate attention to the x-axis scale where we see it does not total 100%. For figure 2 I also recommend adding footnotes that provide more details of the items/resources that are included in the costs for those categories. I have seen other studies provide this as the general perspective is the table should provide enough detail to stand alone.

Authors’ response: Thank you for the comment. We have now changed the figure to a pie chart. However, the item/resources used for the cost category were difficult to add in the footnote, as the supplies and equipment used for laboratory testing are several in numbers. We have included the list of supplies and equipment in the supporting document.

The discussion is written well and informative. My only suggestion it to add more about how Ethiopia can use these results for budgeting and priority planning and speak to the benefit of the study for future economic evaluation.

Authors’ response: Thank you for your comments. We have now included the following in the conclusion section (please see Conclusion section, lines 319-321-309): “The unit cost estimates for COVID-19 sample collection, testing and contact racing could be used as an input by researchers for conducting economic evaluation of COVID-19 interventions. In addition, it may help policy and decision makers for planning and budgeting on future pandemic response and control measures. ”

The references with website links need to be reviewed. I received an error for the links for references #1, 17, and 18 when attempting to open.

Authors’ response: Thank you for pointing out these errors. We have corrected the reference links. Please see reference #1, 17 and 18.

Reviewer #2:

My main concern with the paper is the non-transparency of the data. The costs are clearly communicated, but it is oftentimes unclear where the data comes from. Further, partly because it is unclear where the data come from, it is also unclear whether the results are expected to be representative for Ethiopia as a whole, or just Addis Ababa. And are different components of the total cost estimates more/less representative of Addis Ababa or Ethiopia?

Authors’ response: Thank you for your comment. We have now included a table to provide specific information about the data sources that we have used and to enable replication of our study findings to the greatest extent. Please see supporting document: I. Input and data sources.

The study should be replicable to the greatest extent possible, and right now, the sources of the data are so vague that it would be very hard for someone else to replicate the study. For instance, it says in the first paragraph under the section "Costing approach" that "The data was collected from the EPHI through the review of financial records, literature, and expert consultation." This is very vague -- what data came from what source, and what were the sources more precisely? And, another example: further down in the same section, rental values are discussed, but where does the rental value data come from? It says it was collected from "local experts who engage in activities related to rental services." What does that mean, and how can I find that data?

Authors’ response: Thank you for your comment. We have provided the source of data in the supplementary part of the manuscripts (Supporting document). We also acknowledged the outcome of the study might only be representative for Addis Ababa. To address the comment we have now added the following: “Furthermore, as this study focus in Addis Ababa, we recommend additional studies to be conducted at the national level”. Please see line 3125-317.

---

## [Decision Letter · Decision Letter 1]

11 Apr 2022

PONE-D-21-20233R1Cost-analysis of COVID-19 sample collection, diagnosis, and contact tracing in low resource setting: the case of Addis Ababa, EthiopiaPLOS ONE

Dear Dr. Amanuel Yigezu

Thank you for submitting your manuscript to PLOS ONE. After careful consideration, we feel that it has merit but does not fully meet PLOS ONE’s publication criteria as it currently stands. Therefore, we invite you to submit a revised version of the manuscript that addresses the points raised during the review process.

We look forward to receiving your revised manuscript.

Kind regards,

Carlos Alberto Zúniga-González, Ph.D

Academic Editor

PLOS ONE

Journal Requirements:

Additional Editor Comments:

Dear authors, attend to the recommendations of the reviewer, and I will be waiting for your improvements as soon as possible to continue the process. In general I have reviewed that they have conducted very well the comments of the reviewers.

Reviewers' comments:

Reviewer's Responses to Questions

**Comments to the Author**

1. If the authors have adequately addressed your comments raised in a previous round of review and you feel that this manuscript is now acceptable for publication, you may indicate that here to bypass the “Comments to the Author” section, enter your conflict of interest statement in the “Confidential to Editor” section, and submit your "Accept" recommendation.

Reviewer #1: (No Response)

2. Is the manuscript technically sound, and do the data support the conclusions?

Reviewer #1: Yes

3. Has the statistical analysis been performed appropriately and rigorously? 

Reviewer #1: Yes

4. Have the authors made all data underlying the findings in their manuscript fully available?

Reviewer #1: Yes

5. Is the manuscript presented in an intelligible fashion and written in standard English?

Reviewer #1: No

6. Review Comments to the Author

Reviewer #1: The authors have addressed my previous comments adequately and I recommend publication with the following minor revisions. Recommendations for the authors are to present the cost results in the same order as they are mentioned in the methods and to make sure the abstract method and results are consistent with this as well. This will add clarity in reading the results. Secondly, the authors need to report all costs as they are shown on the figures. For example, personnel costs are separate from food costs in figure 2, so this should be the way it is reported in text. Lastly, there are a few grammatical errors and misspellings throughout this version and I recommend the authors review for grammatical errors and edit as needed prior to publication.

7. PLOS authors have the option to publish the peer review history of their article (what does this mean?). If published, this will include your full peer review and any attached files.

Reviewer #1: No

---

## [Author Response · Author response to Decision Letter 1]

18 May 2022

May 15, 2022

Dr. Sanjay Kumar Singh Patel

Academic Editor

PLOS ONE

Dear Editors,

We are submitting our revised manuscript entitled "Cost-analysis of COVID-19 sample collection, diagnosis, and contact tracing in low resource setting: the case of Addis Ababa, Ethiopia " (PONE-D-21-20233) to PLOS ONE. 

We would like to thank the editor and reviewers for the constructive comments and suggestions which have helped to strengthen our work substantially. 

Below, we have provided a detailed point-by-point response to all comments and suggestions raised in the review.

Thank you very much for your consideration of our manuscript.

Sincerely,

Amanuel Yigezu, on behalf of the authors

 

Point-by-point response to the reviewer’s comments

Reviewer #1: 

The authors have addressed my previous comments adequately and I recommend publication with the following minor revisions. Recommendations for the authors are to present the cost results in the same order as they are mentioned in the methods and to make sure the abstract method and results are consistent with this as well. This will add clarity in reading the results. 

Authors’ response: We thank the reviewer for his/her positive comments on our manuscript. We have now presented the cost results in the same order as they are mentioned in the methods both in the abstract and methods sections. 

Secondly, the authors need to report all costs as they are shown on the figures. For example, personnel costs are separate from food costs in figure 2, so this should be the way it is reported in text. 

Authors’ response: Thank you for the comment. We have now reported food costs in figure 2 the way it is reported in text. 

Lastly, there are a few grammatical errors and misspellings throughout this version and I recommend the authors review for grammatical errors and edit as needed prior to publication

Authors’ response: Thank you for the comment. We have now edited the manuscript to correct grammatical error.

---

## [Editor Report · Decision Letter 2]

23 May 2022

Cost-analysis of COVID-19 sample collection, diagnosis, and contact tracing in low resource setting: the case of Addis Ababa, Ethiopia

PONE-D-21-20233R2

Dear Dr. Amanuel Yigezu,

We’re pleased to inform you that your manuscript has been judged scientifically suitable for publication and will be formally accepted for publication once it meets all outstanding technical requirements.

Kind regards,

Carlos Alberto Zúniga-González, Ph.D

Academic Editor

PLOS ONE

Additional Editor Comments (optional):

Congratulations, all observatios for reviewer have been incorpated.
---

## [Editor Report · Acceptance letter]

30 May 2022

PONE-D-21-20233R2 

Cost-analysis of COVID-19 sample collection, diagnosis, and contact tracing in low resource setting: the case of Addis Ababa, Ethiopia 

Dear Dr. Yigezu:

I'm pleased to inform you that your manuscript has been deemed suitable for publication in PLOS ONE. Congratulations! Your manuscript is now with our production department. 

Kind regards, 

on behalf of

Dr. Prof. Carlos Alberto Zúniga-González 

Academic Editor

PLOS ONE